# Risk factors for blood vessel rupture during vascular access intervention therapy for hemodialysis patients

Yui Kambayashi[1]*, Ken Iseri[2], Tomoki Morikawa[1], Atushi Yao[1], Akio Yokochi[1], Hirokazu Honda[2]

1 Department of Nephrology, Kanto Rosai Hospital, Kawasaki City, Kanagawa, Japan, 2 Department of Nephrology, Showa University School of Medicine, Shinagawa-ku, Tokyo, Japan

* hanamay33@outlook.jp

## Abstract

Blood vessel rupture is a major complication associated with vascular access intervention therapy (VAIVT). However, information regarding the risk factors for ruptures related to VAIVT is limited. The purpose of this study was to investigate the risk factors for rupture during VAIVT. This was a single-center, retrospective observational study. Demographic, clinical, anatomical, and VAIVT procedure variables were reviewed and analyzed using multivariate logistic regression. The 211 patients included in the study underwent 628 VAIVT procedures from November 2019 to December 2021, and 20 blood vessel ruptures occurred. Patients with ruptures had significantly lower BMI (p = 0.043), shorter access vintage(p = 0.017), underwent VAIVT for the first time (p = 0.006), and had lower blood flow quantity (p = 0.005), lower brachial artery flow volume (p = 0.018), and higher resistance index (p = 0.011). The multivariate logistic regression revealed that receiving VAIVT for the first time (OR 5.95, 95%CI 1.01–34.84; p = 0.048) and high resistance index (OR 1.86, 95% CI 1.01–3.16; p = 0.02) were significantly associated with a high risk for rupture. Furthermore, receiver operating characteristic curve analysis to assess the sensitivity-specificity profiles of the resistance index for ruptures showed that the optimal threshold was 0.70 (sensitivity/specificity, 0.69/0.70). Heightened surveillance during vascular access intervention therapy is warranted, especially in patients undergoing VAIVT for the first time or patients with a high resistance index (> 0.70).

## Introduction

Vascular access is a lifeline for adequate hemodialysis therapy [1]. Arteriovenous fistula (AVF) and arteriovenous graft (AVG) are commonly used in clinical practice due to prolonged survival, fewer infections, and lower hospitalization rates [2]. However, the primary 1-year patency rate for AVF and AVG is low due to stenosis or thrombosis (1-year patency rates: AVF 86% and AVG 51%) [3]. A Netherlands study demonstrated that vascular access-related morbidity accounted for 20% of all hospitalizations in hemodialysis patients [4]. Thus, the

**Funding:** The authors received no specific funding for this work.

**Competing interests:** The authors have declared that no competing interests exist.

global standard for maintaining functioning vascular access for hemodialysis is percutaneous transluminal angioplasty, referred to as vascular access intervention therapy (VAIVT). The strengths of VAIVT include an excellent success rate, cost-effectiveness, and reduced hospitalization period compared with surgery [5]. However, VAIVT-related complications include rupture, hematomas, pseudoaneurysm, and thrombosis [6]. Blood vessel ruptures due to vascular wall tears after balloon inflation sometimes require additional hospitalization periods and resources, such as stents [7]. Furthermore, vessel stenosis and patency may be compromised due to ruptures [7–9]. According to previous reports, female, transposed AVF, peripheral cutting balloons, and oversized balloons are significantly associated with an increased risk of rupture [7, 10]. The use of ultrasound for VAIVT is increasing in Japan because the Japanese vascular access guideline recommends ultrasound to eliminate radiation exposure. However, information regarding the association of ultrasound hemodynamic parameters with rupture risk is sparse. Our study aims to investigate the risk factors for blood vessel rupture, especially ultrasound hemodynamic parameters.

## Materials and methods

### Study design and participants

The study protocol, designed as a single-center, retrospective cohort study, was approved by the Kanto Rosai Hospital Ethics Committee. At our hospital, dialysis patients undergo routine ultrasound scans for early detection of stenosis as part of their care. Based on Japanese vascular access guidelines recommending VAIVT if blood quantity flow is under 400ml/min or the resistance index is over 0.6, 211 patients received VAIVT using ultrasound at Kanto Rosai Hospital from November 2019 to December 2021. No patients were excluded. Thus, the analysis was performed on 628 VAIVT procedures performed on 211 patients.

### Definition of exposure and VAIVT procedure

VAIVT was performed based on not only Japanese vascular access guidelines but also in patients who met another clinical criterion, including venous stenosis greater than 50% diagnosed by ultrasound with low blood flow, varix, elevated venous pressures, recirculation, or swelling of the access arm. All procedures were performed under local sedation via puncture of the AVF or AVG and insertion of a 4–6 French sheath depending on the balloon size. Ultrasound (Canon Aplio a Verifia CUS-AA000®) was usually used in our hospital, but digital subtraction angiography was also performed when appropriate. A 0.018-inch or 0.035-inch (TORAY®) guidewire was used, and balloon was chosen among 14 types and the appropriate size of the balloon was selected by matching slightly oversized in comparison to the calibre of the adjacent normal-appearing vessels, then inflated based on our facility's manual. Heparin (1500 units) was administered before inserting the guidewire. The balloon was inflated several times until the indention of the balloon disappeared or reached the maximum rated burst pressure according to the manufacturer's instructions. Blood vessel rupture during VAIVT was diagnosed by Doppler ultrasound. Patients were placed in groups with or without ruptures.

### Clinical parameters

Demographic, clinical, anatomical, blood sample, and vascular access intervention therapy procedure variables were collected. The hemodialysis condition referred to the condition the week before enrollment. Blood samples and X-ray was obtained the month before enrollment.

## Covariates

The following data were collected as covariates: age, sex, body mass index, primary disease of chronic kidney disease, hemodialysis vintage, AVF/AVG, AVF/AVG location, access vintage, operator, location of stenosis, times of receiving VAIVT, blood flow quantity, brachial artery flow volume, resistance index, stenosis diameter, balloon type, sheath size, undergoing VAIVT forwarding or retrograding, maximum inflation pressure, duration of dialysis, the amount body fluid removed, cardio-thoracic ratio, white blood cell count, red blood cell count, mean corpuscular volume, mean corpuscular hemoglobin, hemoglobin, platelet, total protein, albumin, creatinine, blood urea nitrogen (before hemodialysis and after hemodialysis), urea reduction ratio, calcium, phosphorus, intact parathyroid, iron, ferritin, total iron binding capacity, C-reactive protein, sodium, potassium, chloride, and transferrin saturation.

## Statistical analysis

The baseline characteristics are presented as median values (range from 25th to 75th percentiles, IQR) for continuous variables unless otherwise noted and as numbers (percentages) for categorical variables. The two groups were compared using non-parametric Wilcoxon tests for continuous variables and Chi-square tests for nominal variables. Comparisons between three groups were assessed with the non-parametric ANOVA Kruskal Wallis followed by Dunn's test for continuous variables. Non-parametric Spearman rank correlation analysis was used to determine associations between variables. Multivariate logistic regression analysis was used to obtain the odds ratio (OR) for rupture during VAIVT. The following confounders known to impact patency or rupture were selected: age, sex, hemodialysis vintage, primary disease of chronic kidney disease, AVF/AVG, number of VAIVTs, resistance index, type of balloons, and maximum inflation pressure. The receiver operating characteristic (ROC) curves of the resistance index were employed to determine the optimal cutoff values using the Youden index [11].

Statistical significance was set at the level of $P < 0.05$. Missing data were imputed using pairwise deletion. Statistical analyses were conducted using Stata 17 software (StataCorp LLC., College Station, TX).

## Results

### Patient baseline characteristics and clinical parameters

Table 1 shows the baseline characteristics of the 628 VAIVT procedures. The median age was 73 years, and 83% of patients were male. Twenty rupture cases occurred, and two rupture episodes occurred in one patient. When patients were divided into groups with and without ruptures during VAIVT, patients with rupture had significantly lower BMI (p = 0.043), shorter access vintage(p = 0.017), received fewer VAIVT (p = 0.006), and had lower blood flow quantity (p = 0.005), lower brachial artery flow volume (p = 0.018), and higher resistance index (p = 0.011). Among a total of 211 patients in the study, 198 patients had AVF. Table 2 demonstrated the baseline characteristics of patients with AVF.

### Univariate correlations and multivariable predictors of rupture

Surprisingly, univariate Spearman's Rho correlations did not show any significant association between patients with and without ruptures. Table 3 shows the multivariate logistic regression analysis. Receiving vascular access intervention therapy for the first time (OR 5.95, 95%CI 1.01–34.84; p = 0.048) and high resistance index (OR 1.86, 95%CI 1.10–3.16; p = 0.02) were associated with an increased risk of rupture. Fig 1 shows the resistance indices in VAIVT

**Table 1. Baseline clinical and biochemical characteristics of 628 patients receiving vascular access intervention therapy.** (Wilcoxon-test or chi-square test).

| | | Not rupture | Rupture | P-value |
|---|---|---|---|---|
| | | N = 608 | N = 20 | |
| Age(year) | | 73.0 (63.0–79.0) | 73.5(68.5–84.0) | 0.19 |
| sex | Female | 98 (16.1%) | 4 (20.0%) | 0.65 |
| | Male | 509 (83.9%) | 16 (80.0%) | |
| BMI | | 22.7 (20.0–25.2) | 20.7 (18.9–23.3) | 0.043 |
| Primary disease of ESKD | Diabetes mellitus | 329 (54.2%) | 11 (55.0%) | 0.96 |
| | Benign nephrosclerosis | 102 (16.8%) | 4 (20.0%) | |
| | Glomerulonephritis | 97(16.0%) | 2 (10.0%) | |
| | Polycystic kidney disease | 14 (2.3%) | 0 (0.00%) | |
| | Unknown | 653(10.4%) | 3 (15.00%) | |
| Hemodialysis vintage(year) | | 5.0 (2.0–8.0) | 3.5 (1.5–6.5) | 0.18 |
| AVF/AVG | AVF | 534 (88.0%) | 19 (95.0%) | 0.34 |
| | AVG | 73 (11.9%) | 1 (5.0%) | |
| AVF/AVG location | basilic vein-median cubital vein | 1(0.2%) | 0(0%) | 0.55 |
| | brachial artery- basilic vein | 21(3.5%) | 0(0%) | |
| | brachial artery- brachial vein | 22(3.7%) | 0(0%) | |
| | brachial artery- cephalic vein | 9(1.5%) | 0(0%) | |
| | ex AVG-brachial artery | 9(1.5%) | 0(0%) | |
| | ex AVG-brachial vein | 3(0.5%) | 0(0%) | |
| | ex AVG-cephalic vein | 3(0.5%) | 1(5.0%) | |
| | elbow fistula | 32(5.4%) | 0(0%) | |
| | radiocephalic fistula | 478(80.1%) | 18(90.0%) | |
| | AVF in tabaciere region | 1(0.2%) | 0(0%) | |
| | ulnobasilic fistula | 18(3.0%) | 1(5.0%) | |
| Access vintage(year) | | 2(1–5) | 1(1–3) | 0.017 |
| operator | Fellow | 247(41.6%) | 8(40.0%) | 0.89 |
| | Attending | 347(58.4%) | 12(60.0%) | |
| Stenosis location | distal anastomose | 256 (42.2%) | 5 (25.0%) | 0.10 |
| | proximal anastomose | 279 (46.0%) | 14 (70.0%) | |
| | venous outflow | 72 (11.9%) | 1 (5.0%) | |
| Times of VAIVT | 1 | 92 (15.2%) | 8 (40.0%) | 0.006 |
| | 2~4 | 284 (46.8%) | 9(45.0%) | |
| | 5~ | 231(38.1%) | 3 (15.0%) | |
| Qb(ml/min) | | 200.0 (180.0–230.0) | 190.0 (150.0–200.0) | 0.005 |
| FV (ml/min) | | 343.0 (237.0–540.0) | 217.0 (182.5–398.0) | 0.018 |
| RI | | 0.62 (0.53–0.72) | 0.76 (0.59–0.83) | 0.011 |
| Diameter(before:cm) | | 1.0 (0.8–1.4) | 0.9 (0.7–1.3) | 0.23 |
| Types of balloons | Non-compliant balloon catheter | 543 (89.9%) | 19 (95.0%) | 0.45 |
| | Semi-compliant balloon catheter | 61 (10.1%) | 1 (5.0%) | |
| sheath size (Fr) | 4 | 35 (5.8%) | 2 (10.0%) | 0.72 |
| | 5 | 515 (85.0%) | 16 (80.0%) | |
| | 6 | 56 (9.2%) | 2 (10.0%) | |
| forwarding/retrograding | Forwarding | 137 (22.6%) | 3 (15.0%) | 0.42 |
| | Retrograding | 468 (77.4%) | 17 (85.0%) | |
| max inflation atom(atm) | | 18.0 (12.0–24.0) | 17.0 (14.0–24.0) | 0.82 |
| Time of HD(h) | 3 | 20 (3.3%) | 0 (0.00%) | 0.69 |
| | 3~4 | 535 (88.4%) | 18 (90.0%) | |

(*Continued*)

**Table 1.** (Continued)

|  |  | Not rupture | Rupture | P-value |
|---|---|---|---|---|
|  | 4~5 | 50 (8.3%) | 2 (10.0%) |  |
| **The amount of removing body fluid(L)** |  | 2.3 (1.6–3.0) | 2.0 (1.5–2.9) | 0.45 |
| **CTR (%)** |  | 49.5 (45.3–53.4) | 50.5 (48.0–55.2) | 0.20 |
| **WBC($10^3$/μL)** |  | 5.9 (4.8–7.1) | 5.2 (4.5–6.1) | 0.071 |
| **RBC($10^5$/μL)** |  | 349.0 (324.0–380.0) | 347.0 (330.0–386.0) | 0.45 |
| **MCV (fL)** |  | 96.8 (93.0–100.0) | 95.4 (91.4–100.3) | 0.81 |
| **MCH (pg)** |  | 31.1 (29.7–32.4) | 31.4 (28.8–32.2) | 0.77 |
| **Hb(g/dL)** |  | 10.9 (10.2–11.6) | 11.0 (10.4–12.1) | 0.49 |
| **PLT($10^5$/μL)** |  | 17.7 (13.9–21.8) | 16.6 (14.9–18.0) | 0.60 |
| **TP(g/dL)** |  | 6.6 (6.2–7.0) | 6.5 (6.2–6.7) | 0.69 |
| **Alb(g/dL)** |  | 3.5 (3.3–3.7) | 3.4 (3.3–3.8) | 0.86 |
| **Cre(mg/dL)** |  | 10.1 (8.5–12.3) | 8.5 (7.3–11.2) | 0.085 |
| **BUN (before HD: mg/dL)** |  | 62.0 (52.0–73.7) | 63.5 (54.7–73.7) | 0.67 |
| **BUN (after HD: mg/dL)** |  | 19.0 (15.0–24.1) | 20.0 (17.0–25.5) | 0.57 |
| **URR** |  | 69.1 (62.7–73.0) | 67.6 (64.3–71.2) | 0.54 |
| **Calcium(mg/dL)** |  | 8.3 (7.9–8.7) | 8.2 (7.6–8.6) | 0.41 |
| **Phosphorus(mg/dL)** |  | 5.1 (4.3–6.0) | 4.9 (4.4–6.1) | 0.67 |
| **iPTH(pg/mL)** |  | 180.0 (121.0–245.0) | 178.0 (114.0–291.0) | 0.85 |
| **Fe(μg/dL)** |  | 56.0 (43.0–72.0) | 59.5 (43.0–75.0) | 0.89 |
| **TIBC(μg/dL)** |  | 245.0 (214.0–279.0) | 251.5 (234.0–285.0) | 0.42 |
| **ferritin(ng/mL)** |  | 70.3 (39.0–107.5) | 68.3 (39.0–87.7) | 0.84 |
| **CRP (mg/dL)** |  | 0.2 (0.1–0.6) | 0.2 (0.1–0.3) | 0.92 |
| **Na(mEq/L)** |  | 138.0 (136.0–140.0) | 139.0 (138.0–141.0) | 0.15 |
| **K(mEq/L)** |  | 4.6 (4.1–5.2) | 4.7 (4.4–5.2) | 0.44 |
| **Cl(mEq/L)** |  | 102.0 (99.0–104.0) | 102.0 (100.0–105.0) | 0.48 |
| **TSAT (%)** |  | 23.0 (17.2–31.1) | 25.0 (19.0–29.2) | 0.98 |

procedures with and without ruptures. The median resistance index for rupture cases (0.76) was higher than the resistance index for cases with no ruptures (0.62), and the cutoff point was 0.7.

When we reanalyzed the data for patients with AVF only, as a subgroup analysis, the results showed a significant relationship between high resistance index and an increased rupture risk (as demonstrated in Table 4 and Fig 2).

## Optimal cutoff for resistance index for blood vessel ruptures

The optimal cutoff value of the resistance index to predict blood vessel ruptures was determined using ROC curve analysis. The optimal cutoff value and the area under the curve for blood vessel ruptures were 0.70 and 0.69, respectively. The sensitivity and specificity at this cutoff were 0.69 and 0.70, respectively (Fig 3).

We also showed the data for patients with AVF only, the optimal cutoff value and the area under the curve for blood vessel ruptures were 0.70 and 0.72, respectively. The sensitivity and specificity at this cutoff were 0.73 and 0.70, respectively (Fig 4).

## Discussion

Blood vessel rupture is one of the most serious complications of VAIVT, and risk factors for rupture are unclear. Thus, we investigated the risk factors for rupture during VAIVT. Our

**Table 2. Baseline clinical and biochemical characteristics of 553 patients with AVF receiving vascular access intervention therapy.** (Wilcoxon-test or chi-square test).

| | | Not rupture | Rupture | P-value |
|---|---|---|---|---|
| | | N = 534 | N = 19 | |
| Age(year) | | 73.0 (64.0–79.0) | 72.0(68.0–86.0) | 0.28 |
| sex | female | 81(15.2%) | 4 (21.1%) | 0.48 |
| | male | 453 (84.8%) | 15 (78.9%) | |
| BMI | | 22.7(19.9–25.2) | 19.7(18.6–23.6) | 0.038 |
| Primary disease of ESKD | Diabetes mellitus | 300(56.3%) | 11 (57.9%) | 0.90 |
| | Benign nephrosclerosis | 99(18.6%) | 4 (21.1%) | |
| | Glomerulonephritis | 81(13.4%) | 1(5.3%) | |
| | Polycystic kidney disease | 14 (2.6%) | 0 (0.00%) | |
| | Unknown | 49(9.2%) | 3 (15.8%) | |
| Hemodialysis vintage(year) | | 5.0 (2.0–8.0) | 3.0 (1.0–7.0) | 0.26 |
| AVF/AVG location | elbow fistula | 32(6.1%) | 0(0%) | 0.7 |
| | Radiocephalic fistula | 477(90.3%) | 18(94.7%) | |
| | AVF in tabaciere region | 1(0.2%) | 0(0%) | |
| | Ulnobasilic fistula | 18(3.4%) | 1(5.3%) | |
| Access vintage(year) | | 3(1–5) | 1(1–3) | 0.018 |
| operator | fellow | 223(41.8%) | 7(36.8%) | 0.67 |
| | attending | 311(58.2%) | 12(63.2%) | |
| Stenosis location | distal anastomose | 255(47.8%) | 5 (26.3%) | 0.066 |
| | proximal anastomose | 279 (52.2%) | 14 (73.7%) | |
| Times of VAIVT | 1 | 87 (16.3%) | 8 (42.1%) | 0.007 |
| | 2~4 | 233(43.6%) | 8 (42.1%) | |
| | 5~ | 214(40.1%) | 3 (15.8%) | |
| Qb(ml/min) | | 200.0(180.0–230.0) | 180.0 (150.0–200.0) | 0.005 |
| FV (ml/min) | | 349.0 (238.0–539.0) | 213.0 (179.0–361.0) | 0.006 |
| RI | | 0.63(0.54–0.72) | 0.77(0.62–0.84) | 0.005 |
| Diameter(before:cm) | | 1.00 (0.70–1.30) | 0.85 (0.70–1.10) | 0.23 |
| Types of balloons | Non-compliant balloon catheter | 475(89.5%) | 18(94.7%) | 0.46 |
| | Semi-compliant balloon catheter | 56(10.5%) | 1 (5.3%) | |
| sheath size (Fr) | 4 | 35 (6.6%) | 2 (10.5%) | 0.73 |
| | 5 | 454(85.2%) | 15 (78.9%) | |
| | 6 | 44(8.3%) | 2 (10.5%) | |
| forwarding/retrograding | forwarding | 69(13.0%) | 2 (10.5%) | 0.75 |
| | retrograding | 463 (87.0%) | 17 (89.5%) | |
| max inflation atom(atm) | | 18.0 (12.0–24.0) | 18.0 (14.0–24.0) | 0.92 |
| Time of HD(h) | 3 | 15(2.8%) | 0 (0.00%) | 0.66 |
| | 3~4 | 473 (88.9%) | 17 (89.4%) | |
| | 4~5 | 44 (8.3%) | 2 (10.6%) | |
| The amount of removing body fluid(L) | | 2.30(1.7–3.0) | 2.0 (1.4–3.0) | 0.33 |
| CTR (%) | | 49.2(45.1–53.1) | 50.2 (48.0–53.8) | 0.22 |
| WBC($10^3$/μL) | | 5.9(4.8–7.1) | 5.2 (4.5–6.2) | 0.12 |
| RBC($10^5$/μL) | | 349.0(323.0–383.0) | 358.5(329.5–401.5) | 0.39 |
| MCV (fL) | | 97.0(93.3–100.3) | 95.3 (91.2–100.9) | 0.61 |
| MCH (pg) | | 31.2 (29.7–32.6) | 31.5 (28.7–32.2) | 0.62 |
| Hb(g/dL) | | 10.9 (10.3–11.6) | 11.1 (10.3–12.2) | 0.54 |
| PLT($10^5$/μL) | | 18.0(14.4–21.8) | 16.7 (14.6–18.5) | 0.44 |

(*Continued*)

**Table 2.** (Continued)

| | | Not rupture | Rupture | P-value |
|---|---|---|---|---|
| TP(g/dL) | | 6.5(6.2–7.0) | 6.50 (6.2–6.7) | 0.57 |
| Alb(g/dL) | | 3.5 (3.3–3.7) | 3.4 (3.2–3.7) | 0.57 |
| Cre(mg/dL) | | 10.2 (8.4–12.4) | 8.7 (7.3–12.0) | 0.14 |
| BUN (before HD: mg/dL) | | 63.0 (53.2–74.2) | 65.0 (56.4–74.6) | 0.68 |
| BUN (after HD: mg/dL) | | 19.7 (15.6–25.0) | 20.2 (17.9–26.3) | 0.52 |
| URR | | 68.6(62.3–72.7) | 67.5 (62.7–71.0) | 0.47 |
| Calcium(mg/dL) | | 8.3 (7.9–8.7) | 8.2 (7.5–8.6) | 0.36 |
| Phosphorus(mg/dL) | | 5.2(4.4–6.1) | 4.9 (4.5–6.3) | 0.84 |
| iPTH(pg/mL) | | 185.5 (121.5–246.0) | 183.0(127.0–308.0) | 0.72 |
| Fe(μg/dL) | | 58.0 (45.0–74.0) | 63.0 (43.0–75.0) | 0.92 |
| TIBC(μg/dL) | | 247.0 (215.0–277.0) | 255.0 (234.0–285.0) | 0.40 |
| ferritin(ng/mL) | | 69.0 (39.0–106.0) | 67.9 (38.5–85.0) | 0.64 |
| CRP (mg/dL) | | 0.17 (0.06–0.47) | 0.20 (0.09–0.43) | 0.83 |
| Na(mEq/L) | | 138.0 (136.0–140.0) | 139.0 (138.0–141.0) | 0.13 |
| K(mEq/L) | | 4.6 (4.1–5.2) | 4.7 (4.4–5.3) | 0.58 |
| Cl(mEq/L) | | 102.0 (99.0–104.0) | 102.00(100.0–105.5) | 0.53 |
| TSAT (%) | | 23.5 (17.8–32.0) | 26.0 (19.0–29.2) | 0.96 |

Continuous variables are presented as median (25–75 percentile).

Abbreviations: BMI; body mass index, ESKD; end stage kidney disease, AVF; arteriovenous fistula, AVG; arteriovenous graft, VAIVT; vascular access intervention therapy, Qb; quantity of blood flow, FV; flow volume, RI; resistance index, HD; hemodialysis, CTR; cardio-thoracic ratio, WBC; white blood cell, RBC; red blood cell, MCV; mean corpuscular volume, MCH; mean corpuscular hemoglobin, Hb; hemoglobin, PLT; platelet, TP; total protein, Cre; creatinine, BUN; blood urea nitrogen, URR; urea reduction ratio, iPTH; intact parathyroid hormone, TIBC; total iron binding capacity, CRP; C-reactive protein, Na; natrium, K; potassium, Cl; chloride, TSAT; transferrin saturation.

results demonstrated that VAIVT for the first time and a high resistance index were significantly associated with an increased risk for rupture during VAIVT based on multivariate analysis. The ROC curve analysis also revealed that a resistance index over 0.70 predicts rupture.

The mechanism for vascular stenosis is not fully understood. However, local hemodynamic factors, such as asymmetrical blood flow and vascular inflammation, may cause endothelial damage leading to neointimal hyperplasia [4]. We demonstrated a rupture rate during VAIVT of 3.1%, which is in agreement with previously reported rupture rates of 1.7–3.8%; 20 rupture cases occurred during 628 VAIVT procedures. All rupture were diagnosed during VAIVT. No

**Table 3. Predictors of rupture based on multivariate logistic regression analysis.**

| | | OR | Lower 95%CI | Upper 95%CI | p value |
|---|---|---|---|---|---|
| age | | 1.42 | 0.7854222 | 2.579484 | 0.245 |
| sex | | 1.20 | 0.2537452 | 5.715779 | 0.815 |
| Hemodialysis vintage | | 0.96 | 0.8423760 | 1.099986 | 0.576 |
| Primary disease of CKD | | 1.07 | 0.6548059 | 1.752414 | 0.784 |
| AVF/AVG | | 0.62 | 0.0754062 | 5.052233 | 0.653 |
| Times of VAIVT | First time | 5.95 | 1.018903 | 34.84687 | 0.048 |
| | More than second | 3.00 | 0.6072571 | 14.87559 | 0.268 |
| RI | | 1.86 | 1.101986 | 3.163854 | 0.020 |
| Type of balloons | | 0.49 | 0.0604275 | 3.95319 | 0.502 |
| Max inflation atom | | 0.82 | 0.4880628 | 1.386592 | 0.464 |

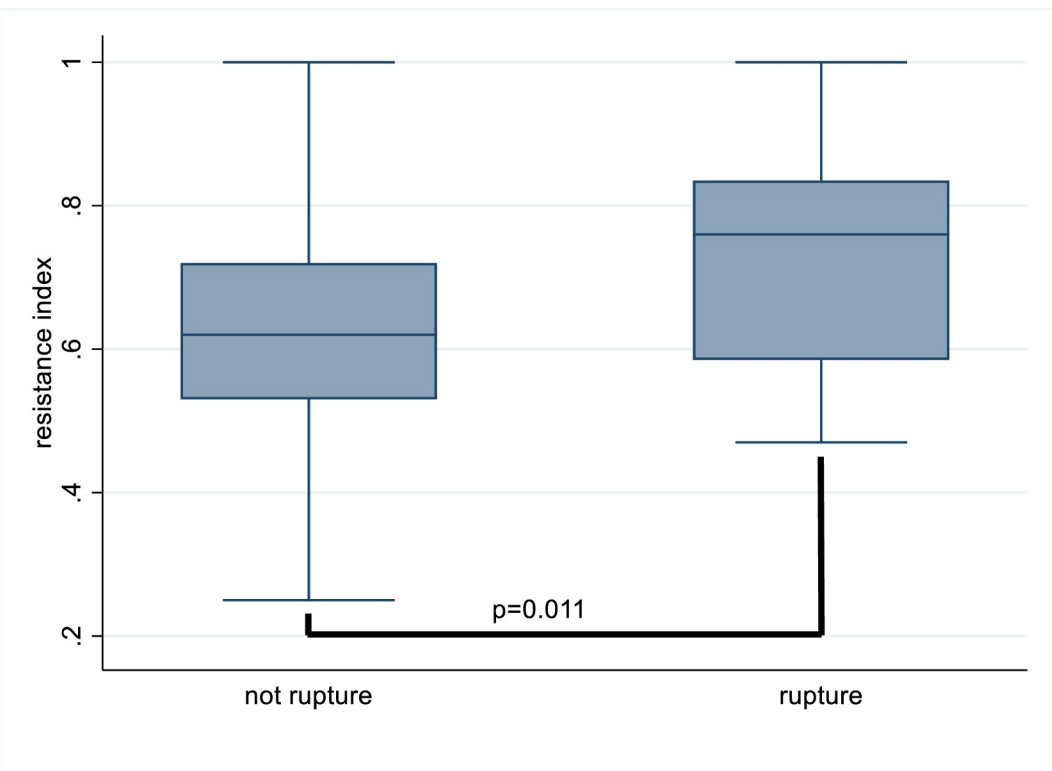

**Fig 1. Relevant to rupture and resistance index.**

patient was diagnosed after leaving the operation room. Among a total of 20 ruptures found in our study, 2 cases were artery ruptures and the other ruptures occurred at the site where the balloon was inflated. All rupture were successfully treated by only balloon tamponade and compression by hand simultaneously at the rupture site. The balloon tamponade was performed at lower pressure for at least 5 minutes but in some cases took time over 15 minutes.

Our study showed that the resistance index reflecting functional lesions was significantly associated with rupture, although no significant association was found between venous stenosis diameters, reflecting morphological lesions, with blood vessel ruptures. Because the resistance index reflects not only venous function but also artery function, the resistance index may be better than venous stenosis diameter for predicting blood vessel rupture [8, 12, 13].

**Table 4. Predictors of rupture based on multivariate logistic regression analysis.** (AVFonly).

| | | OR | Lower 95%CI | Upper 95%CI | p value |
|---|---|---|---|---|---|
| age | | 1.3 | 0.7361999 | 2.463243 | 0.334 |
| sex | | 1.1 | 0.2227429 | 5.338613 | 0.915 |
| Hemodialysis vintage | | 0.97 | 0.8452117 | 1.109971 | 0.646 |
| Primary disease of CKD | | 1.3 | 0.7510335 | 2.100437 | 0.385 |
| Times of VAIVT | First time | 5.7 | 0.9535718 | 33.92763 | 0.056 |
| | More than second | 2.7 | 0.5287467 | 13.93503 | 0.27 |
| RI | | 2.0 | 1.17598 | 3.524161 | 0.011 |
| Type of balloons | | 0.50 | 0.0616867 | 4.123893 | 0.523 |
| Max inflation atom | | 0.85 | 0.5001578 | 1.445295 | 0.549 |

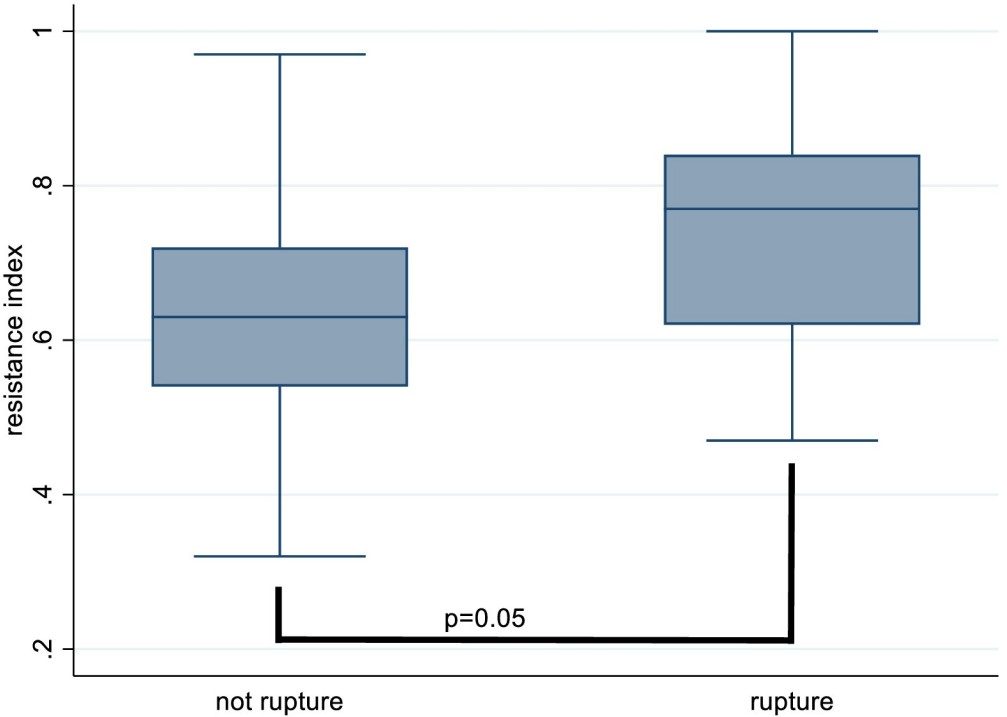

**Fig 2. Relevant to rupture and resistance index.** (AVF only).

According to the guidelines for VAIVT from the Japan and Kidney Disease Outcomes Quality Initiative (KDOQI) [14–16], assessment of the brachial artery resistance index is useful for predicting primary patency rate after creating AVF and for the detection of AVF dysfunction. The optimal resistance index threshold for insufficient blood flow was 0.61 in patients with AVF and 0.57 in patients with AVG [17]. Our results show a higher resistance index associated with an increased risk for blood vessel rupture during VAIVT and may provide important insights into clinical practice, especially when considering VAIVT.

VAIVT for the first time was also significantly associated with an increased risk of blood vessel rupture in our study. To the best of our knowledge, no information regarding the associations between the number of VAIVTs and rupture risks has been reported. However, albeit in a different treatment category, more serious complications were reported after angioplasty for the first time than after the second or third angioplasty during the treatment of coronary artery stenosis [18]. Endothelial injury from forceful dilation of vessels by the balloon, which creates deep cracks in the neointimal tissue leading to inflammation and proliferative response, may cause vessel wall thickening [1, 3, 19]. Vessel wall thickening contributes to the need for repeated VAIVT but might prevent blood vessel rupture during VAIVT.

Surprisingly, none of the factors related to the VAIVT procedures, such as balloon type and inflation pressure, were associated with an increased risk of rupture. A previous study analyzing 1985 VAIVTs showed that female and transposed AVF were significantly associated with an increased risk for rupture [7]. However, we did not detect any significant differences in rupture rates between genders and did not gather information about transposed AVF. Another study analyzing 1242 VAIVTs reported larger balloons (2 mm larger than the normal diameter of the vessels) and peripheral cutting balloons as risk factors for ruptures [10], which was not

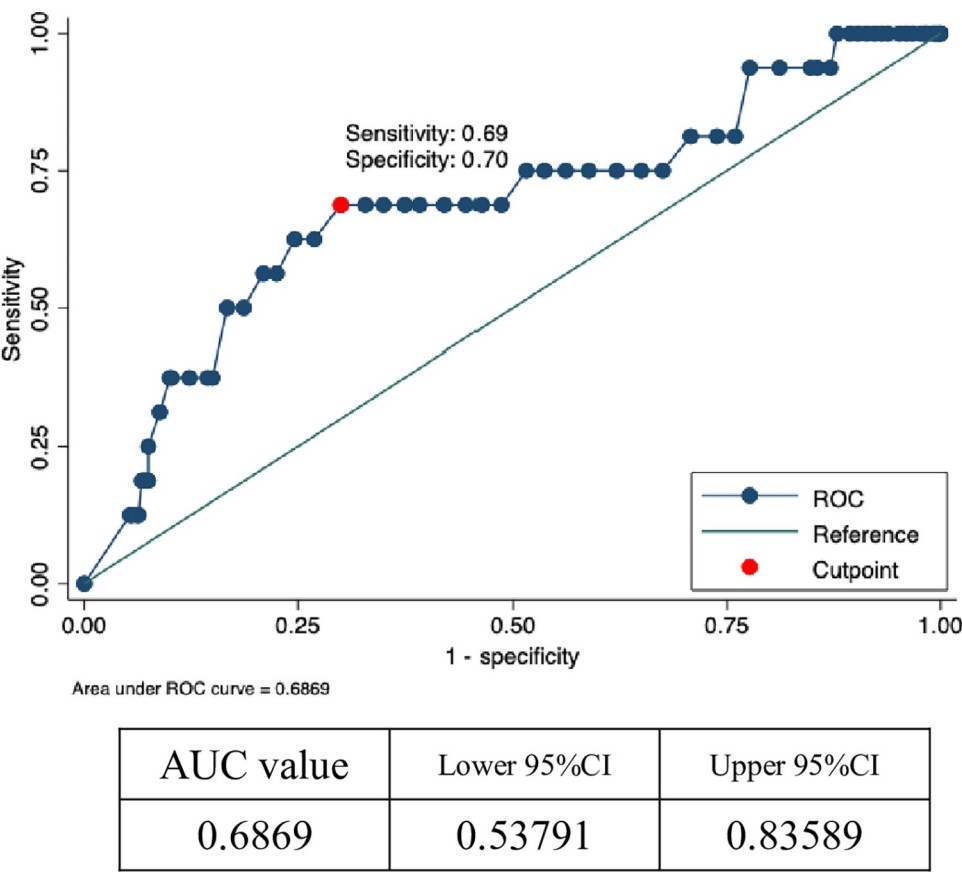

Area under ROC curve = 0.6869

| AUC value | Lower 95%CI | Upper 95%CI |
| --- | --- | --- |
| 0.6869 | 0.53791 | 0.83589 |

**Fig 3. ROC curve analysis of resistance index in patients with rupture.**

in line with our results. This may be due to the different methods for VAIVT. We selected balloon sizes 0–1 mm larger than the normal venous diameter, according to our facility's manual. We usually (over 80%) used non-complaint 5 mm balloons (YOROI5 mm® and MUSTANG5 mm®). Peripheral cutting balloons were not used in this study. The average inflation pressure was 18 atm, and inflation pressures higher than the average did not significantly correlate with rupture risk in our study. Although the procedure for calibrating inflation pressures and the optimal inflation pressures are still under debate, inflating the balloon without high maximum pressure is one of the risk factors associated with the patency rate of AVF [20]. Other studies demonstrated no significant differences in rupture rates between low inflation pressures (10–14 atm) and high inflation pressures (24–30 atm) when dilating central venous stenosis in hemodialysis patients. Considering these facts, clinicians should not hesitate to inflate balloons to high pressures, which may lead to better patency rates [21, 22].

Some limitations to this study should be considered when interpreting the results. First, this was a single-center study, and the sample number was too small. Second, both venous and artery ruptures by guidewire were included. Third, information about transposed AVF, which may be associated with an increased risk for rupture, was not collected [7]. Similarly, peripheral cutting balloons, also known as a risk factor for ruptures, were not used in our facility. Fourth, Qb of 200 ml/min and vascular access flow volume (180–540 ml/min) are common in Japan, but which relatively smaller than those in the western country. These factors might affect the results. Further studies are required to evaluate risk factors for rupture during VAIVT.

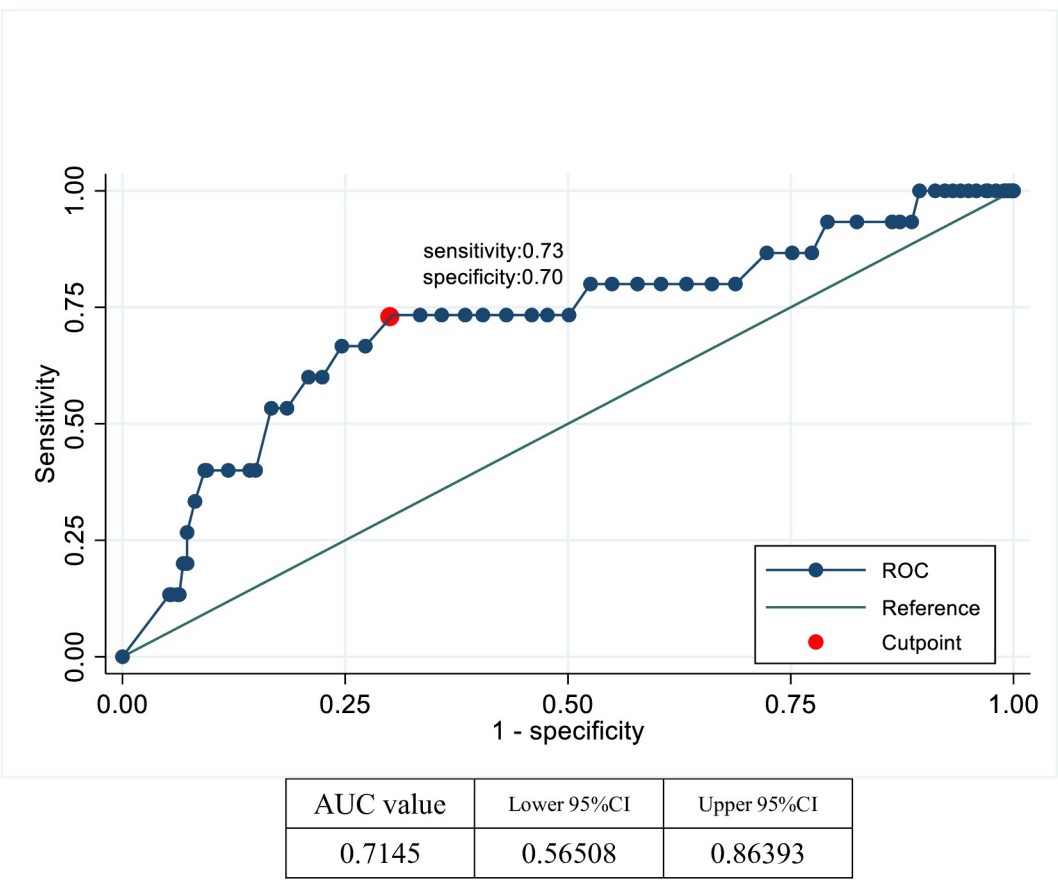

| AUC value | Lower 95%CI | Upper 95%CI |
|-----------|-------------|-------------|
| 0.7145    | 0.56508     | 0.86393     |

**Fig 4. ROC curve analysis of resistance index in patients with rupture.** (AVF only).

## Conclusions

VAIVT is a useful treatment for vascular access failure in hemodialysis patients. Our study demonstrates that high resistance index and VAIVT for the first time were significantly associated with a higher risk of blood vessel ruptures, indicating that increased monitoring and preventative measures may prevent vessel ruptures in patients with these risk factors. However, external validation of these results is needed.

## Supporting information

**S1 File. Participants in research of blood vessels rupture during VAIVT.**
(CSV)

## Acknowledgments

We sincerely appreciate our research team for their help in conducting this research.

## Author Contributions

**Supervision:** Tomoki Morikawa, Atushi Yao, Akio Yokochi, Hirokazu Honda.

**Writing – original draft:** Yui Kambayashi.

**Writing – review & editing:** Ken Iseri.

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
