## [Decision Letter · Decision Letter 0]

31 Jan 2023

PONE-D-22-35142Risk factors for blood vessel rupture during vascular access intervention therapy for hemodialysis patientsPLOS ONE

Dear Dr. Kambyashi,

Thank you for submitting your manuscript to PLOS ONE. After careful consideration, we feel that it has merit but does not fully meet PLOS ONE’s publication criteria as it currently stands. Therefore, we invite you to submit a revised version of the manuscript that addresses the points raised during the review process.

ACADEMIC EDITOR:  Please address the reviewers' comments. 

We look forward to receiving your revised manuscript.

Kind regards,

Tze-Woei Tan, M.D.

Academic Editor

PLOS ONE

Journal Requirements:

"no"

"No authors have competing interests"

6. Please include your tables as part of your main manuscript and remove the individual files. Please note that supplementary tables (should remain/ be uploaded) as separate "supporting information" files

Reviewers' comments:

Reviewer's Responses to Questions

**Comments to the Author**

1. Is the manuscript technically sound, and do the data support the conclusions?

Reviewer #1: Partly

Reviewer #2: Yes

2. Has the statistical analysis been performed appropriately and rigorously? 

Reviewer #1: No

Reviewer #2: Yes

3. Have the authors made all data underlying the findings in their manuscript fully available?

Reviewer #1: Yes

Reviewer #2: Yes

4. Is the manuscript presented in an intelligible fashion and written in standard English?

Reviewer #1: Yes

Reviewer #2: Yes

5. Review Comments to the Author

Reviewer #1: This is a retrospective study analyzing the risk factors for vascular rupture during interventional therapy. This study is trying to answer a very critical clinical question, but the manuscript needs to provide more information

1. How were the study patients selected? What is the referral criteria for VAIVT?

2. Are patients in Japan received routine ultrasound screening or patients is referred when they have issues with their accesses?

3. what is the location of the AVG? how many patients have AVF? Location of the fistula?

4. What is the average vintage of dialysis access?

5. What is the rupture location? Please provide all rupture locations.

6. Fistula and grafts are different accesses. Please separate these two groups when doing analysis.

7. Discussion: “ line 149-150”, “ 20 rupture cases occurred during 627 VAIVT procedures. No ruptures cases occurred after VAIVT in our study”. This sentence is confusing. Can you clarify?

8. What is the diameter differences of angioplasty balloon and the vessel size? Is the differences between angioplasty balloon and vessel diameter different between rupture group and nonrupture group?

9. In my opinion, “first time angioplasty” is not a clear variable to predict rupture. For example, fresh fistula has much higher change to rupture than chronic fistula. Please provide detailed information on access vintage. What is the average access age for nonrupture group compared to ruptured group? Please divide into fistula vs graft group. Please provide the time interval between access creation date and procedure date.

10. Please grade the severity of vessel rupture

11. What is the treatment for vessel rupture in your study? what is the success rate?

11. The experiences of proceduralists correlates strongly with the procedure outcome. Please provide detailed information about procedurlists. Are they fellows or attendings? Ir attendings or vascular surgones etc.

Reviewer #2: This is a well-written manuscript evaluating in a retrospective fashion the risk factors for vessel rupture during dialysis vascular access angioplasty. The authors described their experience in a single-center setting. While the study does have some limitations, as listed in the discussion (relatively small sample size, single center, avoiding cutting-edge balloons), overall it would be a good addition to the current literature.

I would add to the limitations {low Qb of 200 ml/min and vascular access flow volume (180-540 ml/min)}, which perhaps is the standard in Japan, but considered low in other countries.

6. PLOS authors have the option to publish the peer review history of their article (what does this mean?). If published, this will include your full peer review and any attached files.

Reviewer #1: No

Reviewer #2: **Yes: **Khaled Boubes

---

## [Author Response · Author response to Decision Letter 0]

25 Feb 2023

Dear Reviewers

We appreciate reviewers for your precious time in reviewing our paper and providing valuable comments. It was your valuable and insightful comments that led to possible improvements in the current version. The authors have carefully considered the comments and tried our best to address every one of them. We hope the manuscript after careful revisions meet your high standards.

---

## [Decision Letter · Decision Letter 1]

20 Mar 2023

Risk factors for blood vessel rupture during vascular access intervention therapy for hemodialysis patients

PONE-D-22-35142R1

Dear Dr. Kambyashi,

We’re pleased to inform you that your manuscript has been judged scientifically suitable for publication and will be formally accepted for publication once it meets all outstanding technical requirements.

Kind regards,

Tze-Woei Tan, M.D.

Academic Editor

PLOS ONE

Additional Editor Comments (optional):

Thank you for incorporating reviewers' comments in the revision.

Reviewers' comments:

Reviewer's Responses to Questions

**Comments to the Author**

1. If the authors have adequately addressed your comments raised in a previous round of review and you feel that this manuscript is now acceptable for publication, you may indicate that here to bypass the “Comments to the Author” section, enter your conflict of interest statement in the “Confidential to Editor” section, and submit your "Accept" recommendation.

Reviewer #1: All comments have been addressed

Reviewer #2: All comments have been addressed

2. Is the manuscript technically sound, and do the data support the conclusions?

Reviewer #1: Yes

Reviewer #2: Yes

3. Has the statistical analysis been performed appropriately and rigorously? 

Reviewer #1: Yes

Reviewer #2: Yes

4. Have the authors made all data underlying the findings in their manuscript fully available?

Reviewer #1: Yes

Reviewer #2: Yes

5. Is the manuscript presented in an intelligible fashion and written in standard English?

Reviewer #1: Yes

Reviewer #2: Yes

6. Review Comments to the Author

Reviewer #1: The authors addressed the reviewer's comments very well.

The revised manuscript is well written.

The readers would learn new information about vessel rupture after reading the manuscript.

Reviewer #2: This is a good study looking into the risk factors of vessel rupture during interventions on AVF's.. Thanks for addressing the previous concerns.

7. PLOS authors have the option to publish the peer review history of their article (what does this mean?). If published, this will include your full peer review and any attached files.

Reviewer #1: No

Reviewer #2: No

---

## [Editor Report · Acceptance letter]

24 Mar 2023

PONE-D-22-35142R1 

Risk factors for blood vessel rupture during vascular access intervention therapy for hemodialysis patients 

Dear Dr. Kambayashi:

I'm pleased to inform you that your manuscript has been deemed suitable for publication in PLOS ONE. Congratulations! Your manuscript is now with our production department. 

Kind regards, 

on behalf of

Dr. Tze-Woei Tan 

Academic Editor

PLOS ONE